# Ulcerative skin lesions among children in Cameroon: It is not always Yaws

**Jean-Philippe Ndzomo Ngono**[1©], **Serges Tchatchouang**[1©], **Mireille Victorine Noah Tsanga**[1], **Earnest Njih Tabah**[2,3], **Albert Tchualeu**[4], **Kingsley Asiedu**[5], **Lorenzo Giacani**[6], **Sara Eyangoh**[1‡], **Tania Crucitti**[1‡*]

1 Centre Pasteur du Cameroun, Yaounde, Cameroon, 2 National Yaws, Leishmaniasis, Leprosy and Buruli ulcer Control Programme, Ministry of Public Health, Yaounde, Cameroon, 3 Faculty of Medicine and Pharmaceutical Sciences, University of Dschang, West Region, Cameroon, 4 National Laboratory of Public Health, Yaounde, Cameroon, 5 Department of Control of Neglected Tropical Diseases, WHO, Geneva, Switzerland, 6 Department of Medicine, Division of Allergy and Infectious Diseases, and Department of Global Health, University of Washington, Seattle, Washington, United States of America

© These authors contributed equally to this work.
‡ These authors are joint senior authors on this work.
* crucittitania@gmail.com

**Data Availability Statement:** All relevant data are within the manuscript and its supporting information files.

## Abstract

Outbreaks of yaws-like ulcerative skin lesions in children are frequently reported in tropical and sub-tropical countries. The origin of these lesions might be primarily traumatic or infectious; in the latter case, *Treponema pallidum* subspecies *pertenue*, the yaws agent, and *Haemophilus ducreyi*, the agent of chancroid, are two of the pathogens commonly associated with the aetiology of skin ulcers. In this work, we investigated the presence of *T. p. pertenue* and *H. ducreyi* DNA in skin ulcers in children living in yaws-endemic regions in Cameroon.

Skin lesion swabs were collected from children presenting with yaws-suspected skin lesions during three outbreaks, two of which occurred in 2017 and one in 2019. DNA extracted from the swabs was used to amplify three target genes: the human $\beta_2$-microglobulin gene to confirm proper sample collection and DNA extraction, the *polA* gene, highly conserved among all subspecies of *T. pallidum*, and the *hddA* gene of *H. ducreyi*. A fourth target, the *tprL* gene was used to differentiate *T. p. pertenue* from the other agents of human treponematoses in *polA*-positive samples. A total of 112 samples were analysed in this study. One sample, negative for $\beta_2$-microglobulin, was excluded from further analysis. *T. p. pertenue* was only detected in the samples collected during the first 2017 outbreak (12/74, 16.2%). In contrast, *H. ducreyi* DNA could be amplified from samples from all three outbreaks (outbreak 1: 27/74, 36.5%; outbreak 2: 17/24, 70.8%; outbreak 3: 11/13, 84.6%). Our results show that *H. ducreyi* was more frequently associated to skin lesions in the examined children than *T. p. pertenue*, but also that yaws is still present in Cameroon. These findings strongly advocate for a continuous effort to determine the aetiology of ulcerative skin lesions during these recurring outbreaks, and to inform the planned mass treatment campaigns to eliminate yaws in Cameroon.

**Funding:** This work was realised thanks to financial support from WHO and Probitas. Both contributed to the additional purchase of reagents and consumables for the molecular assays employed. The financial sponsors had no role in study design, data collection and analysis, decision to publish, or preparation of the manuscript.

**Competing interests:** The authors have declared that no competing interests exist.

## Author summary

Yaws caused by *Treponema pallidum pertenue* is one of the most prevalent skin ulcer diseases among children in tropical and sub-tropical countries in Africa and the South-Pacific region. In Cameroon, outbreaks of yaws occur among populations living in remote areas where health infrastructure is lacking. The yaws diagnosis is frequently made clinically, even though rapid and simple serological assays were also introduced to confirm active yaws infection. Lately, studies using molecular amplification assays and performed in the South Pacific and Ghana reported that apart from *T. p. pertenue*, *Haemophiluys ducreyi* is also detected in children presenting with yaws-like lesions. This study was performed in the context of the surveillance of yaws in the East and South region of Cameroon. Molecular tools were used to detect and confirm the presence *T. p. pertenue* in samples suspected of yaws and collected during three outbreaks of ulcerative skin lesions among children in Cameroon. In addition, all samples were analysed for *H. ducreyi*. We found that *H. ducreyi* was present in samples from all three outbreaks, but *T. p. pertenue* was only detected among samples collected during the first outbreak. We concluded that yaws was present in Cameroun but that not all outbreaks of yaws-like skin lesions were attributable to *T. p. pertenue* infection.

## Introduction

Outbreaks of ulcerative skin lesions are frequently reported in several countries in the Pacific region, South East Asia, and West and Central Africa [1,2], including Cameroon. These lesions most commonly affect children and young adults in rural and remote communities, and are frequently found on the lower extremities, which are areas often subject to skin injuries and abrasions that might serve as an entry site for bacteria [1–3], such as the one causing yaws [4] and *Haemophilus ducreyi*.

Yaws is a neglected tropical skin disease caused by the spirochete *Treponema pallidum* subspecies *pertenue* that is spread through skin-to-skin contact. *T. p. pertenue* is closely related to the syphilis spirochete, *T. p. pallidum* [1], as these pathogens differ by less than 0.2% of their genome sequences [5]. *T. p. pertenue* causes, similar to *T. p. pallidum*, a multistage disease characterised by an ulcer in the primary stage. Yaws typically starts with the appearance of a papule, mostly found on the lower limbs, evolving to a papilloma and subsequently into an ulcer which will heal over time. The ulcers may occur either as single or multiple, although the latter is more frequent during the second stage of the disease [6]. The treponemes spread through the bloodstream and secondary lesions may appear on different body locations such as face, neck, arms, and on the soles of the feet, with the latter causing a crab-like gait in the patient due to hyperkeratotic pedal plaques and secondary infections. If untreated, following resolution of the early symptoms, the infection will become latent. During latency there are no physical signs and consequently can only be detected by serology. Overall 10% of the untreated infected individuals will progress to the tertiary stage, which is non-infectious but destructive [1]. The bones, joints and soft tissues may be affected and the patients may suffer from irreversible disabilities such as for example chronic periostitis resulting in saber shin or destructive processes leading to the perforation of the palate and nasal septum [1,3].

From 1950 till 2013 a total of 90 countries worldwide reported cases of yaws [2]. In the 1950s Cameroon reported more than 100,000 annual cases [2]. Aiming to eradicate yaws, mass treatment campaigns with benzathine penicillin G were organized by the World Health

Organisation (WHO) in collaboration with the United National Children's Fund between 1952 and 1964 [2,7,8]. As a result, by the end of the 1960s clinical yaws manifestations were no longer observed in Cameroon and the infection was thought to have been eliminated [2,9,10]. However, yaws continued to be present among the indigenous populations living in the tropical forests, and outbreaks were reported in 2007 and 2008 in the Lomié health district located in the East Province of Cameroon [11] (Fig 1). An epidemiological survey carried out in 2009 in the district of Lomié, reported 167 cases of yaws [9]. The number of reported yaws cases in Cameroon declined from 802 in 2010 to 97 in 2013, but increased thereafter from 530 in 2014 to 890 in 2016 [12].

*Haemophilus ducreyi* is a fastidious small Gram-negative coccobacillus. It is known to be a sexually transmitted pathogen causing genital ulcers called chancroid. Although chancroid has almost disappeared globally, interest on *H. ducreyi* has re-emerged as this pathogen is a cause of childhood skin ulcers. This pathogen has in fact been known since 1889 to cause skin lesions. August Ducreyi observed the development of papules and ulcers when purulent material collected from chancroid lesions was inoculated on the skin of patients' forearms [13]. A century later, Quale *et al.* [14] reported a case of chancroid in an HIV infected man who presented multiple lesions on his legs and foot probably resulting from autoinoculation. Since then, *H. ducreyi* has been reported as cause of yaws-like skin ulcers on mainly the lower limbs of children residing in the South Pacific islands and Ghana [15–18].

Although our work focuses on *T. p. pertenue* and *H. ducrey*, it is worth mentioning that other microbes may also cause skin ulcers, such as *Mycobacterium ulcerans* (the agent of Buruli ulcer), as well as anaerobes and Gram-positive cocci causing a tropical ulcer of polymicrobial

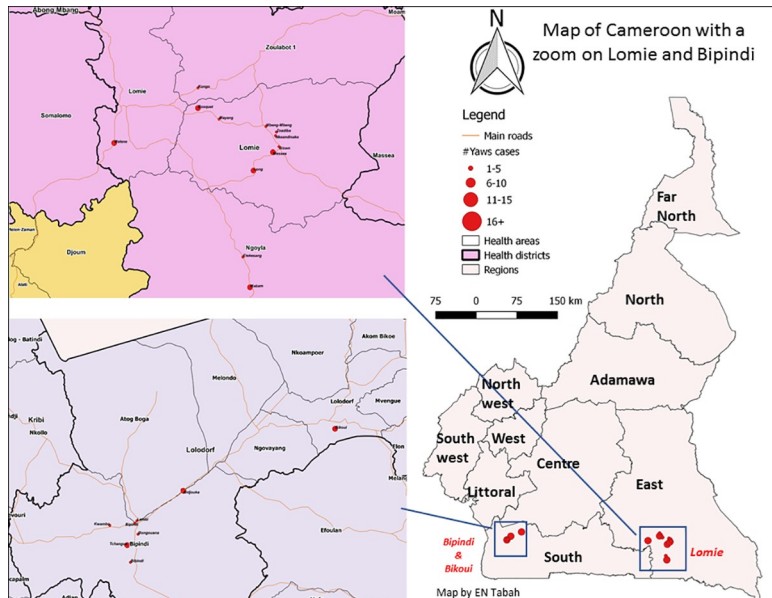

**Fig 1. Geographic location of sampling sites.** Samples were collected from lesions by rotating the sterile tip of the swab gently on the base and centre areas of the lesion. The swab type, storage and transport conditions differed between the epidemiological investigations. During the first outbreak, flocked swabs stored in 1 ml Amies transport medium (Eswab, Copan Diagnostics Inc., Brescia, Italy) were used. In the second outbreak, samples were collected us2ing the Abbott *m*ulti-Collect Specimen Collection Kit (Abbott, Des Plaines, IL, USA) which includes 1.2 mL of specimen transport buffer. During the last investigation, cotton-tipped swabs with wooden shaft (Copan Diagnostics Inc) were used for sample collection and stored dry until use. All samples were transported refrigerated to the Centre Pasteur du Cameroun (CPC) in Yaounde. Samples were stored as received at -20°C until testing. Blood samples were collected for serology to determine the presence of non-treponemal and treponemal antibodies.

etiology. Leishmania species cause cutaneous leishmaniasis which is a non-bacterial skin ulcer. Skin ulcers may also have a non-infectious cause such as in the case of a vascular, a neuropathic, or a traumatic ulcers [19].

To better understand the role played by *T. p. pertenue* and *H. ducreyi* in the aetiology of skin ulcers among children residing in yaws endemic regions in Cameroon, we analysed skin lesion samples collected in the context of three yaws-like outbreaks among children in 2017 and 2019 using nucleic acid amplification tests targeting *T. p. pertenue* and *H. ducreyi*. Our results will inform Public Health officials in Cameroon and other stakeholders in charge of the national surveillance of neglected tropical diseases and the implementation of the national mass drug administration (MDA) programmes that might help reduce the incidence of this serious condition.

## Methods

### Ethics statement

We did not seek additional ethical clearance as the analysis were done in the context of the national skin ulcer lesion surveillance approved by the National Committee on Research Ethics for Human Health (approval reference 2016/08/800/CE/CNERSH/SP). No supplementary data was collected. All parents/ guardians provided oral consent and voluntary opted into the sampling of their children for surveillance purposes. All individuals were managed and treated according to the national Cameroonian guidelines and in line with the WHO recommended guidelines for "Total Targeted Treatment" of yaws [20].

### Sample collection

Samples were collected from skin lesions that occurred in children during two outbreaks in the Lolodorf and Lomie health districts in the South and East regions of Cameroon, respectively, in 2017 and one outbreak in Lolodorf health district 2019 (Fig 1).

### Serology

During the first outbreak, finger prick blood was tested using rapid diagnostic assays. The SD Bioline Syphilis 3.0 (Standard Diagnostics Inc, Suwon, Korea) was used to determine the presence of treponemal antibodies, while the Dual Path Platform (DPP) Screen and Confirm Assay (Chembio Diagnostic Systems Inc, NY, USA) to confirm the presence of an active infection based on the simultaneous detection of both treponemal and non-treponemal antibodies.

Rapid diagnostic tests were not available during the second and third outbreak. Therefore venous blood was collected and tested using the Architect Syphilis TP Assay (Abbott Laboratories, Des Plaines, IL, USA), that detects treponemal antibodies. If positive, the serum was then tested by TPHA (Bio-Rad, Marnes-la-Coquette, France) to confirm the initial result and a non-treponemal test to confirm an active infection (RPR; RPR-nosticon II, BioMérieux, Marcy-l'Etoile, France).

### Nucleic acid amplification

**Sample processing.** Swabs stored dry were thawed at room temperature and biological material was eluted in 500μL lysis buffer (10mM Tris pH 8.0, 0.1M EDTA pH 8.0, 0.5% SDS). All samples were vigorously vortexed for at least 15 seconds and the swabs were pressed against the tube to ensure that most biological material was released in the lysis buffer or transport medium.

**DNA extraction.** DNA was extracted from 200 μL of the samples employing the QIAamp DNA Mini Kit (Qiagen Inc., Valencia, CA) following the manufacturer's protocol for DNA purification from blood and body fluids, with the exception that a) 50 μL proteinase K were added to the samples instead of 20 μL, b) samples were incubated for 1–2 hours at 56°C, and c) 220 μL of AL buffer and 210 μL of 96–100% ethanol were added instead of the suggested 200 μL to compensate for a slightly larger initial sample volume (200 μL instead of 180 μL).

**Polymerase chain reactions.** The presence of host DNA and the absence of amplification inhibitors in the extracted DNA were evaluated by the amplification of a 268-bp fragment of the $\beta_2$-microglobulin gene. The targeted fragment of the *polA* (*tp0105*) gene, conserved present in all subspecies of *T. pallidum* was amplified by real time PCR (qPCR) to detect the presence of *T. pallidum* DNA in the samples. In *polA*–positive samples *T. p. pertenue* DNA was identified based on the amplicon size of the *tprL* target (*tp1031*) which differentiates the yaws pathogen from the other agents of human treponematoses. The amplicon size is +/-209 bp for *T. p. pertenue*, in contrast to the amplicon size of +/-588 bp for *T. p. pallidum*. The amplicon sizes were determined by agarose gel electrophoresis. *H. ducreyi* DNA was detected by amplifying a target of the *hhdA* gene coding for haemolysin. Amplification was performed by qPCR. Primers and probes were synthesized by GenScript, USA. All amplification assays contained positive controls consisting of DNA extracts of *T. pallidum* obtained after culture in a rabbit model and of *H. ducreyi* cultured on agar plates, and a negative control consisting of molecular grade water. In addition, we included environmental DNA checks to control for laboratory area contamination. Each sample was run in duplicate. The PCR methods are summarized in Table 1.

## Definition

An confirmed yaws infection was defined based on the presence of *T. p. pertenue* DNA.

## Data analysis

All data were entered twice in Excel worksheets (see S1 Data) and verified for data entry errors. Simple descriptive statistics were used to summarise patient and sample characteristic, and amplification analysis results. The proportion of boys and girls with *T. p. pertenue* DNA and *H. ducreyi* DNA, respectively, were compared using Fisher's exact tests. The difference in age

**Table 1. Polymerase chain reactions.**

| Target gene | PCR Method | Reaction volume (μL) | DNA volume (μL) | Reagent mixture | Primers/probes (reference) | Cycling conditions | Platform |
|---|---|---|---|---|---|---|---|
| *β2-micro globulin* | End point | 50 | 5 | Go Taq Green Buffer 1X, GoTaq polymerase 0.05U (Promega, USA), 200 μM dNTPs (Invitrogen, USA), 1.5mM MgCl₂ (ThermoScientific, USA) | 320 nM GH20 320 nM PC04 [21] | 3 min 95°C,40x (1 min 95°C, 1 min 60°C, 1 min 72°C)5 min 72°C, hold 15°C | Applied Biosystem geneAmp 9700 |
| *polA* | Real time | 25 | 5 | ABI Taqman FAST Advanced Master Mix (Life Technologies Corporation, USA) | 1.2μM TP1, 1.2μM TP2 180 nM TP3[1] [22] | 2 min 50°C, 30 sec 95°C, 50x (20 sec 95°C, 45 sec 60°C) | ABI PRISM 7500 |
| *tprL* | End point | 50 | 5 | Go Taq Green Buffer 1X, GoTaq polymerase 0.05U (Promega, USA), 200 μM dNTPs (Invitrogen, USA), 1.5mM MgCl₂ (ThermoScientific, USA) | 320 nM TprLpertS 320 nM TprLpert [23] | 5 min 95°C,45x (1 min 95°C, 1 min 60°C, 1 min 72°C)10 min 72°C, hold 15°C | Applied Biosystem geneAmp 9700 |
| *hhdA* | Real time | 25 | 5 | Taqman Universal Master Mix (Life Technologies Corporation, USA) | 400 nM HhdA-F, 600nM HhdA-R 400 nM HhdA-P[1] [24] | 2 min 50°C, 10 min 95°C, 45x (20 sec 95°C, 1 min 60°C) | ABI PRISM 7500 |

[1] The TP3 probe was HEX labelled; HhdA-P probe was JOE labelled.

distribution of children with *T. p. pertenue* DNA and *H. ducreyi* DNA, respectively, were assessed by using a t-test. Significance level was set at 0.05. Analyses were performed in PSPP, an open source version of SPSS.

## Results

### Study population

A total of 114 individuals presenting skin lesions contributed to 112 skin lesion samples: two distinct lesions were sampled in two individuals and lesion samples went missing from four individuals. The distribution of the samples according to the period and location of the outbreak, the age and gender of the individuals and location of the skin lesion is presented in Table 2. The patients' age ranged from 1 to 18 years (median age of 9 years) and 66 children (60%) were male. Of the four individuals from whom skin lesion samples were lacking, three were girls and their age ranged from 8 to 13 years.

### Serodiagnosis

Rapid diagnostic tests were employed for the detection of treponemal and non-treponemal antibodies during the first outbreak. A total of 75 children were tested. Treponemal antibodies were detected by both rapid diagnostic assays, SD bioline and DPP Chembio, in 22/75 (29.3%) individuals. In addition, non-treponemal antibodies were demonstrated in 19 of them.

None of the serum samples collected during outbreak 2 and 3 had a positive serology for treponemal infection; the results are presented in Table 3.

**Table 2. Distribution of the samples according the outbreak's location, age and gender of the individuals contributing to the samples and location of the skin lesions.**

|  | Outbreak 1 | Outbreak 2 | Outbreak 3 |
|---|---|---|---|
| **Period** | September 2017 | December 2017 | August-Octobre 2019 |
| **Location** | Lomie HD | Bipindi, Lolodorf HD | Bikoui, Lolodorf HD |
| **Number of samples** | 75 | 24 | 13* |
| **Number of individuals** | 75 | 24 | 11 |
| **Age Median (IQR)** | 8 (4–11) | 10 (9–11) | 10 (7–15.5) |
| **Gender** |  |  |  |
| **Male** | 49 | 11 | 6 |
| **Female** | 26 | 13 | 5 |
| **Location skin lesion** |  |  |  |
| Lower limb | 53 |  | 9 (foot) |
| Upper limb | 3 |  |  |
| Multiple** | 15 |  |  |
| Trunk | 2 |  |  |
| Head | 1 |  |  |
| Unknown | 1 | 24 | 4 |
| **Type of skin lesion** |  |  |  |
| Ulcer | 60 | NS | 13 |
| Papilloma | 13 |  |  |
| Papule | 1 |  |  |
| Keratosis | 1 |  |  |

* 2 individuals contributed to 2 samples each, both were girls, 15 and 16 years old.

** Multiple: includes lesions on the lower limb; upper limb; among others.

IQR: inter quartile range; HD: health district; NS: not specified.

**Table 3. Presence of treponemal and non-treponemal antibodies in serum samples collected in the context of the three outbreaks.**

| Serology | Outbreak 1 N = 75 (%) | Outbreak 2 N = 24 (%) | Outbreak 3 N* = 7 (%) | Total N = 106 (%) |
|---|---|---|---|---|
| *Treponemal antibodies* | 22 (29.3) | 0 | 0 | 22 (20.8) |
| *Non Treponemal antibodies* | 19 (25.3) | 0 | 0 | 19 (17.9) |

*4 blood samples were not collected.

### Detection of *Treponema pallidum* subsp. *pertenue* and *Haemophilus ducreyi* DNA

One sample out of the 112 samples was excluded from further analysis due to the absence of the amplification of the $\beta_2$-microglobulin gene. The results according to the outbreak are summarized in Table 4.

Treponemal DNA was detected only among the samples collected during outbreak 1 (14/74, 18.9%). *T. p. pertenue* DNA was identified in 12 *polA*-positive samples. *H. ducreyi* DNA was detected in samples collected in all three outbreaks: in 36.5%, 70.8% and 84.6% of the samples collected during outbreak 1, 2, and 3, respectively. Only one sample contained both *T. p. pertenue* and *H. ducreyi* DNA

There was no statistical difference in age distribution of children with confirmed yaws lesions compared to children with *H. ducreyi* lesions. Yaws and *H. ducreyi* lesions were more frequently detected among boys compared to girls but none of the differences were statistically significant (Table 5).

### Serology and presence of *Treponema pallidum* subsp. *pertenue* and *Haemophilus ducreyi* DNA, data of outbreak 1 only

Non-treponemal antibodies, in the presence of treponemal antibodies, were not detected in three children presenting skin lesions in which *T. p. pertenue* DNA was detected and treponemal antibodies were absent in two children with PCR confirmed yaws lesions. On the other hand, *T. p. pertenue* DNA could not be detected in skin lesions of 12 individuals presenting a positive serology (Table 6).

*H. ducreyi* was detected in 27 skin lesion samples. Five *H. ducreyi* infected children had a positive serology for treponemal infection, in one of them *T. p. pertenue* was amplified from the skin lesion and in another *T. pallidum* DNA was detected but *T. p. pertenue* could not be confirmed.

### Discussion

This is the first time that amplification assays to detect *T. p. pertenue* and *H. ducreyi* were employed in Cameroon. Although we were able to confirm the presence of yaws in this

**Table 4. Presence of *T. pallidum*, *T. pallidum pertenue* and *H. ducreyi* DNA in samples collected in the context of the three outbreaks.**

| PCR target | Outbreak 1 N* = 74 (%) | Outbreak 2 N = 24 (%) | Outbreak 3 N = 13 (%) | Total N = 111 (%) |
|---|---|---|---|---|
| *polA* | 14 (18.9) | 0 | 0 | 14 (12.6) |
| *tprL* | 12 (16.2) | NT | NT | 12 (10.8) |
| *hhdA* | 27 (36.5) | 17 (70.8) | 11 (84.6) | 55 (49.6) |

* 1 sample was excluded from further analysis as $\beta_2$-microglobulin could not be detected NT: not tested.

**Table 5. Characteristics of children with PCR confirmed yaws (*T. p. pertenue*) and with *H. ducreyi* lesions.**

|  | Outbreak 1 | Outbreak 2 | Outbreak 3 |
|---|---|---|---|
| **T p pertenue** | N = 12 | N = 0 | N = 0 |
| **Median age (IQR)** | 6 (3.5–9.25) |  |  |
| **Gender** |  |  |  |
| **M (%)** | 9/48 (18.7) |  |  |
| **F (%)** | 3/26 (11.5) |  |  |
| **H ducreyi** | N = 27 | N = 17 | N = 9** |
| **Median age (IQR)*** | 8 (4–10) | 10 (9–11) | 11(9–16) |
| **Gender** |  |  |  |
| **M (%)** | 18/48 (37.5) | 9/11 (81.8) | 5/6 (83.3) |
| **F (%)** | 9/26 (34.6) | 8/13 (61.5) | 4/5 (80) |

*IQR: interquartile range; N: number; M: male; F: female; outbreak 3:

**Each individual was counted once but 2 individuals contributed to 2 samples each, both were girls, 15 and 16 years old. *H. ducreyi* DNA was detected in the four samples.

country, not all yaws-like skin lesions sampled were however attributable to yaws, as *T. p. pertenue* DNA was not detected in the samples collected during the two most recent outbreaks. Conversely, *H. ducreyi* DNA was detected in almost half (49.6%; 55/111) of the samples collected during all three outbreaks.

We found more than twice the number of *H. ducreyi*-positive samples (36.5%) compared to the ones positive for *T. p. pertenue* (16.2%) in the specimens from the first outbreak, which is similar to the findings of a study conducted in Papua New Guinea [17]. In that study, Mitjà *et al*, found almost twice as much of *H. ducreyi* (60%) compared to *T. p. pertenue* (34%) in exudative ulcer material collected from children and young adults [17]. Only *H. ducreyi* DNA was detected in the samples collected during the last two outbreaks. This is in agreement with the results obtained previously in two cross sectional studies performed in the Solomon Islands and Ghana, albeit that we found a much higher proportion of samples with *H. ducreyi* (71% and 85%, detected in outbreak 2 and 3, respectively, compared to 32% and 8% obtained in the Solomon Island and Ghana, respectively) [15,18].

Since the first report of *H. ducreyi* identified in chronic lower limb ulcers in three independent travellers to Samoa [25], evidence of the causative relationship between this pathogen and chronic lower limb ulceration in children and (occasionally) in adults increased significantly [17,18,26,27]. Consequently, nowadays *H. ducreyi* is widely recognized as a causative agent of skin ulcers. However, *H. ducreyi* has also been detected on the skin of asymptomatic children and in environmental samples such as on linen and flies [28]. Therefore, it remains difficult to

**Table 6. Number of samples with positive and negative polA qPCR, tprL PCR and hhdA qPCR results versus serology results, outbreak 1.**

| Serology | *polA* | | *tprL* | | *hhdA* | |
|---|---|---|---|---|---|---|
|  | +<br>N = 14 | -<br>N = 58 | +<br>N = 12 | -<br>N = 2 | +<br>N = 27 | -<br>N = 45 |
| *Treponemal +, Non Treponemal + N = 19* | 8 | 11 | 7 | 1 | 5 | 14 |
| *Treponemal +, Non Treponemal—N = 3* | 3 |  | 3 |  | 0 | 3 |
| *Treponemal–N = 50* | 3 | 47 | 2 | 1 | 22 | 28 |

Note: For what concerns the PCR results of polA and hhdA: a total of 75 samples were collected, however the results presented do not tally due to invalid results in serology or because samples containing inhibitors were not tested by PCR.

distinguish infection from colonisation or contamination. We cannot be certain that *H. ducreyi* is the main etiological cause of the ulcer where this pathogen was detected, rather than a contaminant from the adjacent colonised skin or environment. We can however exclude sample contamination from laboratory sources, as no *H. ducreyi* DNA was present in the environmental controls obtained from laboratory surfaces.

We did not observe significantly more *H. ducreyi* skin lesions among males (49.2%) compared to females (47.7%), which is consistent with previous reports but not with the outcome seen in human infection models [29]. Namely, it was observed that after inoculation of *H. ducreyi* in the skin of human volunteers, men and women formed papules at an equal rate, but that pustules, which erode into ulcers, were twice more frequent among men.

Our results confirm previously reported findings that *H. ducreyi* is frequently present in childhood cutaneous ulcers in yaws endemic, and possible also non-endemic regions. Future research is needed to confirm whether *H. ducreyi* is a pathogen, pathobiont or commensal. At present, we do not know what the findings of skin colonisation and ubiquitously presence of *H. ducreyi* in the environment means in terms of reservoir, infection's risk and ways of transmission.

Yaws is endemic in Cameroon and the low number of *T. p. pertenue* detected in the childhood skin ulcers came as a surprise. Since 2013, the number of reported yaws cases raised consistently until 890 cases in 2016 [12]. We analysed samples collected in 2017 in Lomié, the same region where yaws was reported during 2007–2011, and found *T. p. pertenue* in less than one fifth of the yaws-like lesions. We hypothesise that the number of previously reported yaws cases may be overestimated especially if the diagnosis was based solely on clinical appearance [9].

*T. p. pertenue DNA* was not detected in the clinically suspected yaws ulcerations observed during the most recent two outbreaks. The lack of *T. p. pertenue* in these samples corroborated the non-reactive treponemal serology, indicating that these individuals were not infected with *T. p. pertenue* or other agents of human treponematoses at the time of testing nor were they exposed to treponemal infections in the past. Our results also illustrate how difficult it is to clinically diagnose yaws skin ulcers. In order to improve the accuracy of the diagnosis of yaws, molecular assays should be used for the detection of *T. p. pertenue* in yaws-like lesions.

We could not confirm the diagnosis of yaws by molecular amplification of *T. p. pertenue* DNA target although *T. pallidum* DNA was detected. This is most probably due to differences in the limit of detection of the two amplification techniques. *T. pallidum* DNA is detected using qPCR, whereas the subspecies *pertenue* is identified based on the size of the *tprL* amplicon obtained by qualitative PCR, with the latter method being less sensitive. Consequently, samples with a low concentration of *Treponema pallidum* DNA may be missed when the *tprL* gene is targeted by PCR. However, we do not believe that these two cases were actual syphilis cases. Based on the location of the lesions (legs), the age of the children (10 and 11 years), and the rarity of extragenital, non-sexual transmission of syphilis among children, it is very unlikely that these two cases were due to syphilis infection. However, we cannot exclude that the two children were infected with *T. p.* subspecies *endemicum*, as both bejel and yaws are transmitted by skin-to skin contact and affect similar demographic groups. However, bejel was never previously reported in Cameroun. Furthermore, the localization of the skin lesions does not support the diagnosis of bejel. Ulcers associated with bejel are usually located in the oral cavity or nasopharynx. Whereas the two patients had skin lesions on the lower limb and trunk, which is more typical of yaws infection.

Treponemal and non-treponemal tests are used to confirm active yaws cases in the absence of molecular assays. Active yaws is defined based on the simultaneous presence of non-treponemal and treponemal antibodies, identical to the definition and serological diagnosis of active

syphilis. During the first outbreak, rapid diagnostic tests using finger prick blood were employed. The number of active yaws cases based on reactive serology results (N = 22) was higher than the number of PCR confirmed yaws cases (N = 12). We may explain these findings by the fact that a) the reactive serology may indicate a previously treated or latent yaws infection and not the current cause of the skin lesion; b) the ulcers may be healing and consequently have no detectable *T. p. pertenue*, or c) only one ulcer per individual was sampled even in the presence of multiple skin ulcers, which may have decreased the probability to detect *T. p. pertenue* [15]. On the other hand, five confirmed yaws cases (based on molecular testing) would have been missed using only serology [30]. Three of them were not considered as active yaws based on the lack of reactivity of the non-treponemal test line in the rapid test. This could be due to the lower sensitivity of the DPP test especially for lower RPR titres or to the delayed non-treponemal antibody response observed in early infections [31,32]. It may therefore be recommended to use an automated reader which reports the intensity of the lines in a standardized and objective manner and which has previously be shown to increase the sensitivity of the non-treponemal test line of the DPP RDT [17].

WHO renewed in 2012 its goal to eradicate yaws by 2020 and established a strategy, to meet this target [33]. One of the pillars of this strategy is mass drug administration (MDA) campaigns like the one planned in Cameroon, during which azithromycin will be administered to the entire population of a yaws endemic area. Following MDA, repeat surveys will be organized to identify and treat new cases. Azithromycin is effective against *T. p. pertenue* and *H. ducreyi*, but caution is required as *T. p. pertenue* may develop resistance [34] and *H. ducreyi* may persist on the skin as a member of the skin microbiome. Consequently, reinfection may occur if traumas that compromise the integrity of the skin occur while the subject is no longer protected by antibiotic administration [28]. Indeed, a recent systematic review on the epidemiology of *H. ducreyi* concluded that one round of MDA may not be enough to eradicate the appearance of skin ulcers caused by this pathogen [26]. In addition, recent data about the possible emergence of azithromycin resistance among *H. ducreyi* is lacking [35]. All this considered, MDA may be effective in eradicating of yaws skin ulcers among children, but may overtime be less effective for *H. ducreyi* ulcers which strongly advocates for a continuous surveillance program using molecular diagnostic tools including detection of azithromycin resistance associated mutations.

## Supporting information

**S1 Data. Data collected during outbreak 1, 2, and 3.**
(XLSX)

## Acknowledgments

The authors are indebted to the medical and non-medical staff involved in the surveillance activities of the national Yaws control programme.

## Author Contributions

**Conceptualization:** Jean-Philippe Ndzomo Ngono, Sara Eyangoh, Tania Crucitti.

**Data curation:** Mireille Victorine Noah Tsanga, Earnest Njih Tabah, Albert Tchualeu, Sara Eyangoh.

**Formal analysis:** Jean-Philippe Ndzomo Ngono, Tania Crucitti.

**Funding acquisition:** Kingsley Asiedu, Sara Eyangoh.

**Investigation:** Earnest Njih Tabah, Albert Tchualeu, Sara Eyangoh.

**Methodology:** Earnest Njih Tabah, Kingsley Asiedu, Lorenzo Giacani, Tania Crucitti.

**Project administration:** Sara Eyangoh, Tania Crucitti.

**Resources:** Kingsley Asiedu, Sara Eyangoh.

**Supervision:** Tania Crucitti.

**Validation:** Mireille Victorine Noah Tsanga, Lorenzo Giacani.

**Visualization:** Earnest Njih Tabah.

**Writing – original draft:** Jean-Philippe Ndzomo Ngono, Serges Tchatchouang, Earnest Njih Tabah, Kingsley Asiedu, Lorenzo Giacani, Tania Crucitti.

**Writing – review & editing:** Jean-Philippe Ndzomo Ngono, Serges Tchatchouang, Mireille Victorine Noah Tsanga, Earnest Njih Tabah, Albert Tchualeu, Kingsley Asiedu, Lorenzo Giacani, Sara Eyangoh, Tania Crucitti.

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
