## [Decision Letter · Decision Letter 0]

20 Nov 2020

Dear Dr. Crucitti,

Thank you very much for submitting your manuscript "Ulcerative skin lesions among children in Cameroon: It is not always Yaws." for consideration at PLOS Neglected Tropical Diseases. As with all papers reviewed by the journal, your manuscript was reviewed by members of the editorial board and by several independent reviewers. In light of the reviews (below this email), we would like to invite the resubmission of a significantly-revised version that takes into account the reviewers' comments. 

We cannot make any decision about publication until we have seen the revised manuscript and your response to the reviewers' comments. Your revised manuscript is also likely to be sent to reviewers for further evaluation.

Sincerely,

Melissa J. Caimano

Deputy Editor

Melissa Caimano

Deputy Editor

Reviewer's Responses to Questions

**Key Review Criteria Required for Acceptance?**

**Methods**

-Are the objectives of the study clearly articulated with a clear testable hypothesis stated?

-Is the study design appropriate to address the stated objectives?

-Is the population clearly described and appropriate for the hypothesis being tested?

-Is the sample size sufficient to ensure adequate power to address the hypothesis being tested?

-Were correct statistical analysis used to support conclusions?

-Are there concerns about ethical or regulatory requirements being met?

Reviewer #1: (No Response)

Reviewer #2: The objectives of this study are not clearly spelt out. While in lines 92-93 of the main text, the authors state that they intended to "better understand the aetiology of skin ulcers among children residing in yaws endemic regions in Cameroon," the same indicate in the abstract (lines 6 and 7) that they "investigated the presence of T. p. pertenue and H. ducreyi DNA in skin ulcers in children living in yaws-endemic regions in Cameroon." The latter is more specific and though narrower in its scope, is more applicable to the methodology described. The study design as described cannot address the stated objectives in lines 92-93 of the main text. The authors should be conscious of the broad aetiology of skin ulcers in tropical/sub-tropical areas which include Yaws but also other common differentials such as tropical ulcers of mixed aetiology and other bacterial lesions such as ecthyma. 

Given the understanding that this study was performed within the context of an outbreak investigation, for which reason ethical clearance was not sought, a lot more will be expected of the authors to clearly describe the aetiology of the ulcers encountered by the Public Health Surveillance actors. Going with the broad objective(s) identified within the main text, the authors have not also incorporated the full process of clinical decision making involving some element of clinical history (e.g. evolution of ulcers) and examination (e.g. categorization of ulcers based on appearance - superficial, deep, sloping edge, etc) in their text - these should be included graphically or in tabular forms within manuscript. The educated reader will want to know more about the types of ulcers being tested; while a gold standard diagnostic test greatly supports clinical decision making, the former on its own does not replace the value of good clinical decision making which can have up to an 80% yield from just clinical history. Also, and consequently, the paper is largely silent on the aetiology of skin-ulcers not attributable to H ducreyi and or T. p. pertenue and the influence of trauma which is highlighted by the authors.

Some thoughtful questions to be considered by authors:

- If only one ulcer per individual was sampled even in the presence of multiple ulcers, which the authors state may have reduced the probability of detecting T. p. pertenue and hence the disparity between reactive serological results and PCR confirmation, what was done to resolve this disparity? Of ethical concern too is what was done with those study subjects in question? In the case of outbreak 1, 35 individuals neither had H. Ducreyi or T. p. pertenue confirmed in lesions - what then did they have, and what happened to them? Although that may argued as being in the context of the National Surveillance, it is relevant to make the conclusions and recommendations from this paper stronger.

In lines 205-208, the authors make statements about the differences in age distribution and gender distribution for both sets of PCR confirmed cases of interest, which are not supported within text. The authors should show provide more details about the statistical analysis and results (probably within the table where they present the results).

Reviewer #3: Line 124, The beginning of the sentence should start with a capital letter. Correct is ‘If positive’.

Line 139, A period is required at the end of the sentence.

Line 302-305, I agree that the emergence of azithromycin resistance mutations is very important in the countermeasures of T.p. pertenue. But why aren't your samples tested for resistance mutations?

I recommend you to detect resistance mutations of azithromycin in your sample.

**Results**

-Does the analysis presented match the analysis plan?

-Are the results clearly and completely presented?

-Are the figures (Tables, Images) of sufficient quality for clarity?

Reviewer #1: (No Response)

Reviewer #2: The results are clearly presented within the manuscript, but the details of statistical analysis as indicated in previous section, should be added. A suggestion is also made to the authors, to add a clearer map detailing where cases were found within the Lomie, Bikoe and Bipindi districts.

Reviewer #3: Line 225-227, These data were not found in the figures or tables. Where are these shown?

Line 227, It was written that “T. pallidum but not T. p. pertenue was detected (from skin lesion samples of a child)”, which is very important. Because there are not so many reports of Nonvenereal Transmission of Syphilis, (J Williams et. al, Indian J Sex Transm Dis. 1990;11(1):27-8 and Pingyu Zhou et. al, Sexually Transmittsd Diseases 2009;36(4):216-217, etc.) you should discuss this case. Also, this case may be venereal, as the report by Elise Klouman et al. suggests (Genitourin Med 1997;73:522-527). In addition, Bejel, another nonvenereal syphilis, has been reported in warm, moist areas, which is not the arid areas, unlike traditional wisdom, so this point may also need to be considered.

**Conclusions**

-Are the conclusions supported by the data presented?

-Are the limitations of analysis clearly described?

-Do the authors discuss how these data can be helpful to advance our understanding of the topic under study?

-Is public health relevance addressed?

Reviewer #1: (No Response)

Reviewer #2: The conclusions of this study are of public health relevance and support similar evidence from the Solomon Islands and Ghana, as indicated by the authors. While the data presented by study adds to the body of knowledge supporting the need for better characterization and diagnosis of skin ulcers in tropical/sub-tropical areas, addressing the issues about the methodology will further enhance its advancement of our understanding of this topic. Clearly there are still several questions unanswered, even from their own clearly spelt out limitations of the analysis. 

The statement in lines 279 to 281 - "Our results also illustrate how difficult it is to clinically diagnose yaws in skin ulcers" may be right but the subsequent statement "and that molecular assays should be used to determine the aetiology of the lesion" may be a logical jump that is not supported by the data or the analysis, given the limitations spelt out and the challenges regarding the methodology to answer the broad objective identified in lines 92-93. 

While the authors' call for a continuous surveillance program after MDA's is justified, again the data given the limitations and challenges with methodology, do not support restricting the nature to just the use of molecular diagnostic tools. It makes sense from their data presented, that post MDA, surveillance should involve both serological and molecular methods, fine-tuned to meet operational constraints.

Reviewer #3: Line 234, A period is required at the end of the sentence.

Line 302-305, I agree that the emergence of azithromycin resistance mutations is very important in the countermeasures of T.p. pertenue. But why aren't your samples tested for resistance mutations? (Reposting of Methods comment.)

Line 296, Citation [29,30] should be [28,29].

Line 301, Citation [28] should be [30].

Line 306, Citation [29] should be [31]

**Editorial and Data Presentation Modifications?**

Reviewer #1: (No Response)

Reviewer #2: Statements in lines 63-64 and line 68 should be supported by in-text citations.

Reviewer #3: (No Response)

**Summary and General Comments**

Reviewer #1: Line 84 - I think it is clear that H Ducreyi definitely causes skin ulcers so I would remove the word possible here. Equally in the discussion I think there is considerably less doubt about its causative role than is presented here - it is isolated from lesions, clearly causes ulcers in an experimental model, the lesions heal in response to targeted treatment of ducreyi. We find staph aureus on intact skin but it is also clear that it definitely causes skin lesions so I am not sure the fact that we find ducreyi on intact skin really alters the overall strong evidence that kochs postulates have been fufilled. 

1) The information on sero-diagnosis should be presented much much earlier in the results as this is key. 

For example given that 0 cases in outbreak 2 & 3 were associated with positive serology the fact that these cases was not yaws is completely unsurprising. 

- Serology breakdown should be included in the tables related to the outbreaks.

2) Related to this Figure 2 is difficult to follow.

I would suggest a table showing

Percent dual positive (Trep/non-Trep), percent trep only, percent both negative - and then the PCR breakdown across these groups. 

3) I presume all the lesions were ulcerative without papillomas? Ulcers have a hugely wide differential and it is not at all surprising that most are not caused by T.p.pertenue (simialr results for example are found when people investigate possible Buruli ulcers) 

4) No information is given about what reporting in Cameroon is based on but I presume as with elsewhere that it is clinical reporting without any serological confirmation. If so then it should be made clear that national reporting data is likely to be highly unreliable. 

5) Lines 293 - I would suggest referencing the syphilis literature as it is very clear (and unsurprising) that primary lesions may be PCR positive before seroconversion and therefore before a DPP or similar assay will be positive. 

6) The raw data required to reproduce the analysis are not available with the submission. PLease provide these.

Reviewer #2: Overall a useful paper which is well written, but will greatly benefit from a review of the methodology to meets its broad objective and enhance its value in furthering our knowledge on the aetiology of skin-ulcers in tropical/sub-tropical areas. While the authors indicate that ethical clearance was not sought since this was in the context of the national skin ulcer surveillance and no supplementary data was collected, given that this secondary data analysis identifies people in living specific districts (they may be of a particular tribe for example), the issues of potential harm to subjects and return for consent need to be considered by an ethical review board, even if it is not a complete review. This will strengthen this publication and the conclusions from it.

Reviewer #3: Line 306, Macrolide Resistance of T.p. pertenue was described in 2015 by David Šmajs et. aｌ. in Am. J. Trop. Med. Hyg., 93(4):678-683. So, citation should be changed.

PLOS authors have the option to publish the peer review history of their article (what does this mean?). If published, this will include your full peer review and any attached files.

Reviewer #1: No

Reviewer #2: No

Reviewer #3: Yes: Takuya Kawahata, Osaka Institute of Public Health, Japan
---

## [Editor Report · Decision Letter 1]

26 Jan 2021

Dear Dr. Crucitti,

We are pleased to inform you that your manuscript 'Ulcerative skin lesions among children in Cameroon: It is not always Yaws.' has been provisionally accepted for publication in PLOS Neglected Tropical Diseases.

Best regards,

Carlos Franco-Paredes

Associate Editor

Melissa Caimano

Deputy Editor

---

## [Editor Report · Acceptance letter]

10 Feb 2021

Dear Dr. Crucitti,

We are delighted to inform you that your manuscript, "Ulcerative skin lesions among children in Cameroon: It is not always Yaws.," has been formally accepted for publication in PLOS Neglected Tropical Diseases.

Best regards,

Shaden Kamhawi

co-Editor-in-Chief

Paul Brindley

co-Editor-in-Chief
